# Metabolic Remodeling with Hepatosteatosis Induced Vascular Oxidative Stress in Hepatic ERK2 Deficiency Mice with High Fat Diets

**DOI:** 10.3390/ijms23158521

**Published:** 2022-07-31

**Authors:** Takehiko Kujiraoka, Kazuki Kagami, Toyokazu Kimura, Yuki Ishinoda, Yasunaga Shiraishi, Yasuo Ido, Shogo Endo, Yasushi Satoh, Takeshi Adachi

**Affiliations:** 1Division of Cardiovascular Medicine, National Defense Medical College, 3-2 Namiki, Tokorozawa 359-8513, Japan; kujiraniku5221@hotmail.co.jp (T.K.); mirror.1028k@gmail.com (K.K.); oyotikuuyakim@gmail.com (T.K.); isshiishigoto@yahoo.co.jp (Y.I.); sirayasu10@hotmail.co.jp (Y.S.); yido@bu.edu (Y.I.); 2Aging Regulation Research Team, Tokyo Metropolitan Geriatric Hospital and Institute of Gerontology, 35-2 Sakaecho, Tokyo 173-0015, Japan; sendo@tmig.or.jp; 3Department of Anesthesiology, National Defense Medical College, 3-2 Namiki, Tokorozawa 359-8513, Japan; ys@ndmc.ac.jp

**Keywords:** ERK2, metabolic syndrome, metabolism, hepatosteatosis, insulin resistance, endothelial dysfunction

## Abstract

We previously demonstrated the marked hepatosteatosis and endothelial dysfunction in hepatocyte-specific ERK2 knockout mice (LE2KO) with a high-fat/high-sucrose diet (HFHSD), but detailed metabolic changes and the characteristics in insulin-sensitive organs were not tested. This study aimed to characterize metabolic remodeling with changes in insulin-sensitive organs, which could induce endothelial dysfunction in HFHSD-LE2KO. The serum glucose and fatty acid (FA) were modestly higher in HFHSD-LE2KO than HFHSD-Control. FA synthesis genes were up-regulated, which was associated with the decreased phosphorylation of AMPK and ACC, and with the up-regulation of SREBP-1 in the liver from HFHSD-LE2KO. In FA and amino acids fraction analysis, arachidonic acid/eicosapentaenoic acid ratio, L-ornithine/arginine ratio, asymmetric dimethylarginine and homocysteine levels were elevated in HFHSD-LE2KO. Insulin-induced phosphorylation of AKT was blunted in skeletal muscle. Serum leptin and IL-1β were elevated, and serum adiponectin was decreased with the enlargement of epididymal adipocytes. Finally, the enhanced superoxide levels in the aorta, which were blunted with CCCP, apocynin, and tempol, were observed in HFHSD-LE2KO. A pre-incubation of aortic rings with tempol improved endothelial dysfunction in HFHSD-LE2KO. HFHSD-LE2KO revealed an acceleration of FA synthesis in the liver leading to insulin resistance in skeletal muscle and the enlargement of visceral adipocytes. Global metabolic remodeling such as changes in arginine metabolism, ω3/ω6 ratio, and adipocytokines, could affect the vascular oxidative stress and endothelial dysfunction in HFHSD-LE2KO.

## 1. Introduction

Insulin resistance is a major characteristic of metabolic syndrome (MetS) and type 2 diabetes mellitus (T2DM), leading to cardiovascular complications [1]. Hepatosteatosis (HST) is characterized almost universally by insulin resistance and is strongly associated with T2DM and obesity. HST is associated with moderately increased cardiovascular events among T2DM individuals, independently of classical risk factors [2,3,4,5,6]. We have revealed that HST was closely associated with endothelial dysfunction [7,8]. The liver plays a central role in lipid, carbohydrate, and amino acid metabolisms, which could affect insulin resistance and vascular oxidative stress.

Insulin signaling comprises two major cascades, the insulin receptor substrate (IRS)/phosphatidylinositol 3′-kinase (PI3K)/protein kinase B (AKT) pathway and the Ras/Raf/mitogen-activated protein kinase (MEK)/extracellular signal-regulated kinase (ERK) pathway. The role of the hepatic IRS/PI3K/AKT pathway in insulin signaling has been extensively studied, mainly in vitro. AKT regulates intracellular metabolism such as glucose/amino acid transport, glycogen synthesis, lipid synthesis, and protein synthesis [9]. Kahn’s group studied hepatocyte-specific insulin receptor-deficient mice and noted the impairment of gluconeogenesis and hyperlipidemia associated with atherosclerosis susceptibility [10,11]. On the other hand, the Ras/Raf/MEK/ERK pathway was also activated with many hormones, vasoactive substrates, and adipocytokines in MetS/T2DM. However, the tissue-specific role of ERK in insulin resistance has not been fully clarified.

Previously, we created hepatocyte-specific ERK2 deficient mice (LE2KO) and fed them with a high-fat/high-sucrose diet (HFHSD) [7]. HFHSD-LE2KO revealed marked deterioration in HST and insulin resistance, which was associated with aortic oxidative stress and endothelial dysfunction. Endoplasmic reticulum (ER) stress with the decreased sarco/endoplasmic reticulum Ca^2+^-ATPase 2 (SERCA2) expression in the liver from HFHSD-LE2KO and serum glucose and fatty acid (FA) modestly increased; however, they might not simply explain the cause of aortic oxidative stress and endothelial dysfunction. In the previous report, we partially assessed the changes in metabolism and did not show the pathological changes in insulin-sensitive organs, such as skeletal muscle and visceral adipose tissue. From these backgrounds, we evaluated the other mechanism of FA synthesis in the liver, insulin sensitivity in skeletal muscle, and the size of adipocytes in visceral fats. Moreover, we analyzed the systemic metabolic changes, including FA/amino acid fractions and adipocytokines to seek the factors inducing oxidative stress and endothelial dysfunction in HFHSD-LE2KO. 

## 2. Results

### 2.1. The Aggravation of HST without Changing Body Weight in HFHSD-LE2KO

LE2KO and control littermates were fed with either HFHSD or normal chow (NC) for 20 weeks. As we reported before [7], LE2KO showed similar increases in body weight with HFHSD for 20 weeks (NC-Control 31.4 ± 0.7 g, NC-LE2KO 31.0 ± 0.8 g, HFHSD-Control 41.5 ± 0.9 g *, HFHSD-LE2KO 41.6 ± 0.9 g *, *n* = 8, * *p* < 0.05 vs. NC-Control) (Figure 1A). However, HFHSD-LE2KO significantly increased liver weight and fat deposition compared with HFHSD-Control (Figure 1B,C). The weights of other organs were similar in HFHSD-Control and HFHSD-LE2KO (Appendix A). Serum alanine aminotransferase levels and hepatic triglyceride contents were markedly higher in HFHSD-LE2KO compared with HFHSD-Control, as shown in the previous report [7].

### 2.2. Decreased Phosphorylation of AMPK/ACC and the Increased Expression of SREBP-1c in Liver from HFHSD-LE2KO

We employed the pathway analysis of gene chip and compared the gene expression of the liver from HFHSD-LE2KO and HFHSD-Control in the previous report [7,8] and the Appendix A. The enzymes for FA synthesis were up-regulated, and the enzyme for FA oxidation was down-regulated in HFHSD-LE2KO. AMP-activated protein kinase (AMPK) is the potent regulator for the balance of FA oxidation/synthesis. Immuno-blotting of hepatic homogenate showed the decreased phosphorylation of AMPK and acetyl-CoA carboxylase (ACC), a downstream enzyme of AMPK (Figure 2A–C). The expression of the sterol regulatory element-binding transcription factor 1c (SREBP-1c), which is the master gene for the transcription of FA synthesis, was markedly increased in HFHSD-LE2KO by reverse transcription-polymerase chain reaction (RT-PCR) (Figure 2D). We did not find the significant upregulation of genes for gluconeogenesis, such as glucose6-phosphatase or phosphoenolpyruvate carboxykinase (data not shown).

### 2.3. Elevation of Serum Glucose and Insulin Levels and Insulin Resistance in Skeletal Muscle from HFHSD-LE2KO

Blood fasted and fed glucose levels and serum insulin levels were higher in HFHSD- Control compared with NC-Control and were higher in HFHSD-LE2KO (Fast: NC-Control 77.3 ± 7.0 mg/dL, NC-LE2KO 76.0 ± 6.4 mg/dL, HFHSD-Control 117.2 ± 6.3 mg/dL, HFHSD-LE2KO 149.7 ± 10.3 mg/dL *^#^, Fed: NC-Control 138.4 ± 12.4 mg/dL, NC-LE2KO 147.8 ± 8.6 mg/dL, HFHSD-Control 190.1 ± 5.5 mg/dL, HFHSD-LE2KO 230.0 ± 10.5 mg/dL *^#^, Serum insulin: NC-Control 0.48 ± 0.07 ng/mL/dL, NC-LE2KO 0.50 ± 0.15 ng/mL/dL, HFHSD-Control 1.05 ± 0.18 ng/mL/dL, HFHSD-LE2KO 1.74 ± 0.21 ng/mL/dL l *^#^, *n* = 8, * *p* < 0.05 vs. NC-Control, ^#^ *p* < 0.05 vs. HFHSD-Control). As mentioned above, the expression of gluconeogenesis genes in the liver was not significantly different between HFHSD-LE2KO and HFHSD-Control; the insulin sensitivity in the skeletal muscle was tested with insulin-induced phosphorylation of AKT in the skeletal muscle at 15 min [12]. HFHSD-Control revealed a marked decrease in pAKT/total AKT by insulin compared with NC-Control, and HFHSD-LE2KO further decreased it compared with HFHSD-Control (Figure 3). These data suggested that the deterioration in insulin resistance in the skeletal muscle could increase blood glucose by the reduced glucose uptake in HFHSD-LE2KO.

### 2.4. Adipocytes Enlargement and Changes in Serum Adipocytokines in Epididymal Fat

The weights of epididymal and perirenal fat were increased with HFHSD; however, there was no difference between HFHSD-Control and HFHSD-LE2KO (Appendix A). The size of adipocytes was larger in HFHSD-LE2KO than HFHSD-Control (Figure 4A,B). Since the enlargement of visceral fat might modulate adipocytokine releases, we assessed the serum adiponectin, leptin, and IL-1β levels. Serum adiponectin levels were lower, and serum leptin and IL-1β levels were higher in HFHSD-LE2KO than HFHSD-Control (Figure 4C–E).

### 2.5. Serum FA Fraction and Amino Acid Analysis

Because the FA synthesis pathway was upregulated with the decreased phosphorylation of AMPK and ACC, serum lipids were measured. There was no significant difference in serum total cholesterol, triglyceride, and high-density lipoprotein (HDL)-cholesterol between HFHSD-Control and HFHSD-LE2KO. Serum FA was modestly higher in HFHSD-LE2KO (NC-Control 0.66 ± 0.03 mEq/L, NC-LE2KO 0.67 ± 0.03 mEq/L, HFHSD-Control 0.73 ± 0.03 mEq/L, HFHSD-LE2KO 0.84 ± 0.03 mEq/L *^#^, *n* = 8, * *p* < 0.05 vs. NC-Control, ^#^ *p* < 0.05 vs. HFHSD-Control). We further assessed the serum FA fraction with gas chromatography-mass spectrometry (GC-MS) (Table 1). A total of 6 of 24 FA were increased (Oleic acid (18:1 ω9), 5-8-11 Eicosatetraenoic acid (20:4 ω6), Dihomo-γ-linolenic acid (20:3 ω6), Arachidonic acid (AA: 20:4 ω6), Behenic acid (22:0), and Nervonic acid (24:1 ω9) and 2 of 24 FA were decreased (α-Linolenic acid (18:3 ω3) and Eicosapentaenoic acid (EPA: 20:5 ω3) in HFHSD-Control compared with NC-Control. A total of 9 of 24 FA were higher concentration (Stearic acid (18:0), Arachidic acid (20:0), 5-8-11 Eicosatetraenoic acid (20:4 ω6), Dihomo-γ-linolenic acid (20:3 ω6), AA (20:4 ω6), Erucic acid (22:1 ω9), Docosatetraenoic (22:4 ω6) acid, Docosahexaenoic acid (DHA: 22:6 ω3), and Nervonic acid (24:1 ω9)) and 5 of 24 FFA were lower concentration (Myristic acid (14:0), Palmitoleic acid (16:1), γ-Linolenic acid (18:3 ω6), α-Linolenic acid (18:3 ω3), and EPA (20:5 ω3)) in serum from HFHSD-LE2KO than from HFHSD-Control. The EPA/AA ratio, regarded as a cardiovascular risk for the deficiency of ω3 FA, decreased with HFHSD and was further reduced in HFHSD-LE2KO compared with HFHSD-Control. 

### 2.6. Serum Amino Acid Analysis

From these results, HFHSD-LE2KO revealed broad changes in serum metabolites potentially related to cardiovascular diseases. Thus, we also performed the serum amino acid (AA) analysis (Appendix A and Figure 5). First, we measured serum homocysteine and asymmetric dimethylarginine (ADMA) by Enzyme-Linked Immuno Sorbent Assay (ELISA). Serum homocysteine and ADMA levels were higher in HFHSD-LE2KO than HFHSD-Control. We further assessed the serum AA fraction with liquid chromatography-tandem mass spectrometry (LC-MS). Only 1 in 27 AA was increased (3-Methylhistidine), and 7 in 24 AA were decreased (Threonine, Valine, Isoleucine, Leucine Phenylalanine, Lysine, Arginine) in HFHSD-Control compared with NC-Control. A total of 7 in 27 AA were higher (Glutamic acid, α-Aminoadipic acid, Citrulline, Tyrosine, Monoethanolamine, Tryptophan, Ornithine), and 1 in 27 AA was lower (α-Aminobutyric acid) in HFHSD-LE2KO than in HFHSD-Control. HFHSD-LE2KO increased in ornithine/arginine ratio and decreased in GABR (Arginine/Citrulline + Ornithine) compared with HFHFD-Control.

### 2.7. HFHSD-LE2KO Demonstrated Increased Aortic Superoxide Production and Impaired Endothelium-Dependent Relaxation

No remarkable histological differences in cardiovascular morphology were observed between HFHSD-LE2KO and HFHSD-Control. We next assessed aortic superoxide levels with dihydroergotamine (DHE) staining. Aortic superoxide levels were elevated by about two-fold in HFHSD-LE2KO compared with HFHSD-Control. Pre-incubation of aortic rings with CCCP (2 μM, 60 min), apocynin (100 μM, 60 min), and tempol (100 μM, 60 min) blunted the fluorescence intensity of DHE staining in HFHSD-LE2KO (Figure 6A,B).

The isometric tension measurement assessed vascular relaxation in aortic rings. Acetylcholine (ACh)-induced relaxation (endothelial-dependent relaxation [EDR]) was markedly impaired in HFHSD-LE2KO compared with HFHSD-Control. A pre-incubation of the aorta with tempol (100 μM, 60 min) markedly improved EDR in HFHSD-LE2KO, resulting in no differences in EDR between HFHSD-LE2KO and HFHSD-Control with tempol (Figure 6C). Exogenous nitric oxide from sodium nitroprusside (SNP) -induced relaxation was identical with or without tempol (Figure 6D).

## 3. Discussion

Obese-related diseases such as MetS/T2DM, characterized by insulin resistance, are important risk factors for cardiovascular diseases. Insulin resistance is closely associated with HST, although how HST affects cardiovascular complications is controversial. There were reports that HST was associated with hypertension, endothelial dysfunction, and heart failure in patients [13]. Previously, we reported that HFHSD-LE2KO revealed the marked progression of HST without changing body weight, serum total cholesterol, and triglyceride levels. HFHSD-LE2KO revealed hepatic ER stress with the decreased SERCA2 expression, which might be related to FA synthesis in the liver. HFHSD-LE2KO increased aortic superoxide levels, which might be caused by NOX1 and NOX4 upregulation, and the endothelial dysfunction caused by the decreased eNOS phosphorylation at Ser1179 [7]. HFHSD-LE2KO is an exciting model to investigate how HST with insulin resistance leads to endothelial dysfunction; however, important points remain to be clarified in the previous manuscript [7]. (1) The mechanism of the accelerated lipid synthesis in the liver was relatively superficially shown. (2) The functional changes in other insulin target organs (skeletal muscle, visceral adipose tissue) in HFHSD-LE2KO were not tested. (3) Serum glucose, FA, and insulin levels modestly increased in HFHSD-LE2KO. Although the very high glucose could induce endothelial dysfunction (>500 mg/dL) [14], the increase in serum glucose was very modest in HFHSD-LE2KO. Additional factors were required for the marked endothelial dysfunction in HFHSD-LE2KO. From these backgrounds, we further analyzed the metabolic changes in HFHSD-LE2KO compared with HFHSD-Control. We found accelerated FA synthesis with decreased phosphorylation of AMPK and upregulation of SREBP-1c in the liver. Progression of insulin resistance in skeletal muscle and the enlargement of adipocytes in epididymal fats were observed. Changes in FA and amino acid fractions, and serum adipocytokine levels were detected. We considered that these multiple factors with metabolic remodeling could potentially induce aortic oxidative stress and endothelial dysfunction in HFHSD-LE2KO. 

Hepatic insulin resistance is characterized by upregulation of lipid synthesis and gluconeogenesis [15]. In the pathway analysis, HFHSD-LE2KO showed the upregulation of lipid synthesis and the downregulation of lipid oxidation genes. AMPK is an important molecule to regulate the balance between lipid syntheses. The activation of AMPK phosphorylated to inhibit ACC. ACC produces malonyl-CoA from acetyl-CoA, and malonyl-CoA is a precursor to induce lipid synthesis. Malonyl-CoA is also an allosteric inhibitor for carnitine -O-palmitoyl transferase to inhibit FA oxidation [16]. Thus, the decreased phosphorylations of AMPK/ACC were suggested to induce FA synthesis and suppress FA oxidation in HFHSD-LE2KO. The expression of SREBP-1c, a master transcription factor for FFA, was markedly upregulated in HFHSD-LE2KO [17]. These data suggested that HFHSD-LE2KO promoted lipid synthesis in the liver and worsened HST by inhibiting AMPK and upregulation of SREBP-1c. 

We also assessed insulin-induced pAKT in the skeletal muscle. The phosphorylation and the activation of AKT trans-located glucose transporter 4 and increased glucose uptake to the skeletal muscle resulting in decreased blood glucose. Insulin-induced pAKT in skeletal muscle was reduced in HFHSD-Control and further decreased in HFHSD-LE2KO. Thus, the decreased insulin sensitivity in the skeletal muscle might partially explain the increased blood glucose and insulin levels in HFHSD-LE2KO. High glucose, FA, hormones, and inflammation can induce insulin resistance in the skeletal muscle by inhibiting IRS1 [18,19]. Because we found various metabolites and hormones to induce insulin resistance in HFHSD-LE2KO, the metabolic remodeling in the liver might secondarily cause insulin resistance in the skeletal muscle. 

The accelerated lipid synthesis of the liver and the increased serum FA levels could cause the enlargement of visceral adipose tissue. We did not observe the changes in weight of the epididymal and peri-renal fat; however, microscopic analysis showed the enlargement of adipocytes in epididymal fat in HFHSD-LE2KO. Visceral fat is an endocrine organ that releases adipocytokines. We found decreased adiponectin and increased leptin, typically observed in the MetS model with the enlargement in visceral fat [20]. IL-1β, an important cytokine to induce adipose tissue inflammation [21], also increased in HFHSD-LE2KO. These secondary changes in adipocytokines may contribute to the various complications of HST with insulin resistance, including vascular oxidative stress.

A serum FA fraction was analyzed to assess the metabolic remodeling in HFHSD-LE2KO. ω6 and ω9 FA and saturated Behenic acid (22:0) were increased in HFHSD-Control compared with NC-Control. On the other hand, 2 of ω3 FA (α-Linolenic acid (18:3 ω3) and EPA20:5 ω3) were decreased in HFHSD-Control. In HFHSD-LE2KO, saturated FA (Stearic acid (18:0), Arachidic acid (20:0)), ω6 FA, ω9 FA, and DHA were increased. A-Linolenic acid (18:3 ω3), and EPA (20:5 ω3), which were decreased in HFHSD-Control, were further reduced in HFHSD-LE2KO. ω6 FAs are metabolized to AA, a precursor of prostanoids, and associated with inflammation [22]. Major products of prostanoids such as thromboxane (TX) A2 and prostaglandin (PG) H2 activate the TP receptor and impair NO bioactivity [23]. ω3 FA is metabolized to EPA. The prostanoids generated from EPA, such as PGH3 and TXA3, can compete with the TP receptor, suppressing the inflammation and maintaining the NO bioactivity [24]. Changes in the balance between ω6/ω3 FA may contribute to the decreased NO bioactivity to promote endothelial dysfunction in HFHSD-LE2KO. Myristic acid (14:0) and palmitoleic acid (16:1) are also reduced in HFHSD-LE2KO compared with HFHSD-Control. Palmitoleic acid was found to be rich in the liver, and the administration of palmitoleic acid was reported to improve insulin resistance in mice [25]. The decreased palmitoleic acid may also contribute to insulin resistance in HFHSD-LE2KO. 

Amino acid analysis characterized metabolic remodeling in HFHSD-LE2KO. Serum ADMA, an intrinsic NO synthase inhibitor, was increased in HFHSD-LE2KO. ADMA was known as the biomarker of cardiovascular risk in the Framingham study [26]. Homocysteine, another biomarker for cardiovascular risk, is also higher in HFHSD-LE2KO than HFHSD-Control [27]. HFHSD-Control decreased branched-chain amino acids and glutamic acids compared with NC-Control, which are the risks of diabetes [26,27]. Interestingly, changes in arginine metabolism were prominent in HSHD-LE2KO. Serum L-Citrulin and L-Ornithine were increased in HFHSD-LE2KO. The activation of the urea cycle and inducible NO synthase could increase these metabolites [28]. Increases in the ornithine/arginine ratio or a decrease in GABR (Arg/Cit+Orn) were associated with cardiovascular events [29,30], which were observed in HFHSD-LE2KO. The deterioration in HST and abnormal arginine metabolism may contribute to cardiovascular risk and diseases.

HST in HFHSD-LE2KO raised multiple factors to promote endothelial dysfunction, such as high glucose, hyperinsulinemia, and high FA levels. The decreased adiponectin and the increased leptin and inflammatory cytokines may also lead to endothelial dysfunction. The decreased EPA/AA ratio is a risk for atherosclerosis [31]. The elevated ADMA, ornithine/arginine ratio, and homocysteine levels can also lead to endothelial dysfunction. Multiple factors in the HST of HFHSD-LE2KO may contribute to cardiovascular risk. The high superoxide levels reduce NO bioactivity, a known mechanism for endothelial dysfunction. The previous manuscript shows the upregulation of NOX1 and NOX4 [7]. In this study, we pharmacologically tested the source of the aortic superoxide in HFHSD-LE2KO. Apocynin and CCCP blunted DHE staining of the aorta, suggesting NOX and mitochondrial electric-transport chains may be major sources of superoxide. Aortic rings were pre-incubated with tempol, a superoxide scavenger, to test the theory that superoxide production was a major cause of endothelial dysfunction. Tempol blunted DHE staining and markedly improved endothelial dysfunction in HFHSD-LE2KO. Thus, metabolic remodeling with progressed HST in HFHSD-LE2KO induced endothelial dysfunction mainly via superoxide productions. The study may raise the targets of metabolites to improve endothelial dysfunction and cardiovascular complications in MetS/T2DM with HST. 

## 4. Materials and Methods

### 4.1. Animals, Genotyping, and Diets

ERK2 floxed mice were generated as previously described [7,32]. They were backcrossed with C57BL/6J for more than ten generations. We crossed these mice with albumin promoter-driven Cre transgenic mice, which were maintained on the same background (C57BL/6J). To ablate ERK2 selectively in the liver, we targeted ERK2 using the Cre-loxP strategy. A floxed allele in which loxP sites flanked exons 2 and 3 of ERK2 was constructed as reported previously [33]. An albumin promoter-driven Cre transgenic mouse line (Alb-Cre), in which Cre activity is confined to the liver, was used to drive recombination. The resulting Alb-Cre^(+/−)^; ERK2^(lox/lox)^ mice (also called LE2KO), and the littermate controls (Alb-Cre^(−/−)^; ERK2^(lox/lox)^ mice) were used in this study [7]. 

Animals were maintained in a temperature-controlled facility on a 12-h light/12-h dark cycle (lights on from 7:00 am. to 7:00 pm.). They were fed a normal rodent diet containing 4.6% crude fat with less than 0.02% cholesterol (Clea Japan, Inc., Tokyo, Japan) ad libitum. Eight-week-old male LE2KO and control littermates were fed with either NC or HFHSD with percentages of calories from carbohydrates of 28.3%, fat of 54.5%, and protein of 17.2%, (Oriental Yeast, Suita, Japan) for 20 weeks. Genotyping for the ERK2 floxed allele and presence of Cre was performed by PCR analysis using genomic DNA isolated from the tail tip. 

### 4.2. Biochemical Analysis of Tissues

All samples were homogenized as described previously [7]. Tissues were removed and homogenized with a homogenization buffer (20 mM Tris-HCl [pH 7.4], 150 mM NaCl, 1 mM Na_2_EDTA, 1 mM EGTA, 1% NP-40, 2.5 mM sodium pyrophosphate, 1 mM β-glycerophosphate, 1 mM Na_2_VO_4_) containing 1mM PMSF and a protease inhibitor cocktail. The homogenates were centrifuged at 13,000× *g* for 20 min at 4 °C. Protein concentrations were measured by the Bradford assay using bovine serum albumin as a standard [34]. Protein lysates were resolved by SDS-PAGE and transferred to PVDF membranes at a voltage of 30 V for 2 h at 4 °C, and immunoblotted with primary antibodies to AMPK, phosphor-AMPK (Thr172), ACC, phosphor-ACC (Ser 79), AKT, phosphor-AKT (Ser 473), β-actin and GAPDH. (Cell Signaling Technology, Danvers, MA, USA). 

### 4.3. Tissue Preparation and Histology

Mice were euthanized by pentobarbital injection and perfused with 0.9% saline followed by 4% paraformaldehyde. The liver and epididymal fat were fixed in 10% formalin for 24 h, embedded in paraffin, and sectioned. All samples were routinely stained with hematoxylin and eosin staining. 

### 4.4. Measurements of Metabolites and Adipocytokines

Blood glucose levels were determined using a glucose detecting kit (Wako Pure Chemical Industries, Osaka, Japan). Serum insulin (Mercodia, Sylveniusgatan, Uppsala Sweden), adiponectin (Otsuka, Japan), leptin (R&D System, Minneapolis, MN, USA), and interleukin (IL) −1β (RayBiotech, Peachtree Corners, GA, USA) levels were measured using ELISA kits. Serum levels of TC, TG, HDL-C, and FFA were assessed by enzymatic assays (Wako Pure Chemical Industries, Osaka, Japan). Serum homocysteine and ADMA levels were determined by high-performance liquid chromatography with fluorescent detections [30]. The FA fraction was measured with GC-MS, and amino acid analysis was measured with LC-MS by SRL (Tokyo, Japan) [35]. 

### 4.5. Detection of Insulin-Induced p-AKT in Skeletal Muscle

Mice were anesthetized, and soleus muscles appeared by cutting skins. A total of 0.75 U/kg body weight of human regular insulin (Humulin R, Eli Lilly, Indianapolis, IN, USA) was injected into the caudal vein. After 15 min of injection, soleus muscles were cut and rapidly frozen with liquid nitrogen. pAKT was detected with immunoblotting.

### 4.6. Quantitative Real-Time PCR for SREBP1c

Total RNA was isolated from the livers using TRI reagent (Sigma-Aldrich, St. Louis, MO, USA). Complementary DNA was synthesized using SuperScript III reverse transcriptase (Invitrogen, Carlsbad, CA, USA) according to the manufacturer’s protocol. Quantitative mRNA expression was assessed by real-time PCR with Power SYBR Green PCR Master Mix (Applied Biosystems, Foster City, CA, USA). Samples were duplicated on the ABI PRISM 7700 (Applied Biosystems, Foster City, CA, USA). The following oligonucleotide primer pairs were used: SREBP1c (forward), 5′-TGAGAAGCGCTACCGGTCTT-3′;SREBP1c (reverse), 5′-AAGCGGATGTAGTCGATGGC-3′;18S rRNA (forward), 5′-TTCCGATAACGAACGAGACTCT-3′;18S rRNA (reverse), 5′-TGGCTGAACGCCACTTGTC-3′.

### 4.7. Gene Microarray Analysis

For the Oligo-DNA microarray analysis between livers from LE2KO and controls fed HFHSD, the “3D-Gene” Mouse Oligo chip 24k (Toray Industries Inc., Tokyo, Japan), which contains 23,522 distinct genes, was used. Total RNA was labeled with Cy3 or Cy5 using the Amino Allyl Message AMP II aRNA Amplification Kit (Applied Biosystems). The Cy3- or Cy5-labeled aRNA samples were pooled in a hybridization buffer, and hybridized for 16 h. The hybridization was performed using the supplied protocols (www.3d-gene.com (accessed on 28 July 2022)). Pathway analysis was performed by Toray (Japan).

### 4.8. Measurement of Vascular Superoxide Production

The superoxide production of aortic rings was assessed with DHE (Invitrogen Molecular Probes, Eugene, OR, USA), as previously described. [36] Frozen sections of the aortic rings were immediately cut into 10-μm-thick sections and placed on glass slides. Samples were then incubated at room temperature for 30 min with DHE (2 × 10^−6^ M) and protected from light. Images were obtained with a microscopic system (BZ-8000; Keyence, Japan) with an excitation wavelength of 540 nm and an emission wavelength of 605 nm. The fluorescence intensity of DHE staining was measured using NIH Image J 1.42 (National Institutes of Health, public domain software). Some aortic rings were pre-incubated with CCCP (2 μM, 60 min), apocynin (100 μM, 60 min), and tempol (100 μM, 60 min). 

### 4.9. Preparation of Aortic Rings and Organ Chamber Experiments

Measurement of isometric tension was performed as described previously [37]. Thoracic aortas from mice were cut into 3-mm rings with special care taken to preserve the endothelium and mounted in organ baths filled with Krebs–Ringer bicarbonate solution (NaCl 118.3 mM, KCl 4.7 mM, CaCl_2_ 2.5 mM, MgSO_4_ 1.2 mM, KH_2_PO_4_ 1.2 mM, NaHCO_3_ 25 mM, D-glucose 5.5 mM) aerated with 95% O_2_ and 5% CO_2_ at 37 °C. The preparations were attached to a force transducer, and the isometric tension was recorded. Vessel rings were primed with KCl (30 mM) and then precontracted with L-phenylephrine (10^−6.5^ M), producing a submaximal contraction. After the plateau was attained, the rings were exposed to increasing concentrations of ACh (10^−9^ to 10^−5^ M) or SNP (10^−9^ to 10^−5^ M) to obtain cumulative concentration-response curves. Some of the rings were pre-incubated with tempol (100 μM, 60 min).

### 4.10. Statistical Analysis

Results are presented as mean ± SEM or SD. Data for vascular relaxation were analyzed by a two-way ANOVA with repeated measures followed by the post hoc test with a Bonferroni correction for multiple comparisons. The other data were analyzed by one-way ANOVA followed by the post hoc test with a Bonferroni correction for multiple comparisons. All statistical analyses were performed with the GraphPad Prism Software Ver. 5.02 (San Diego, CA, USA). A *p* value of less than 0.05 was considered to be statistically significant. In the graphed data, *, ** and *** denote *p* values of <0.05, 0.01, 0.001, respectively.

## Figures and Tables

**Figure 1 ijms-23-08521-f001:**
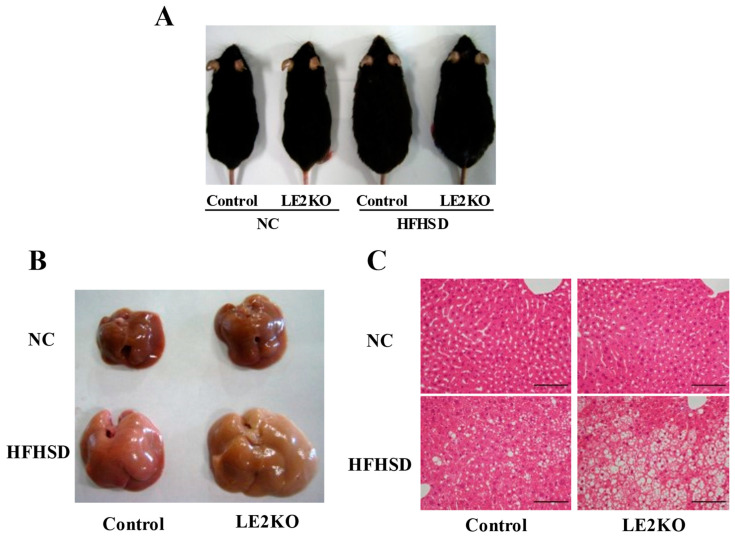
Typical body appearance and macroscopic or microscopic images of liver. LE2KO showed similar increases in body weight with HFHSD for 20 weeks (**A**) HFHSD-LE2KO significant increased liver weight (**B**) and fat deposition (**C**) compared with HFHSD-Control.

**Figure 2 ijms-23-08521-f002:**
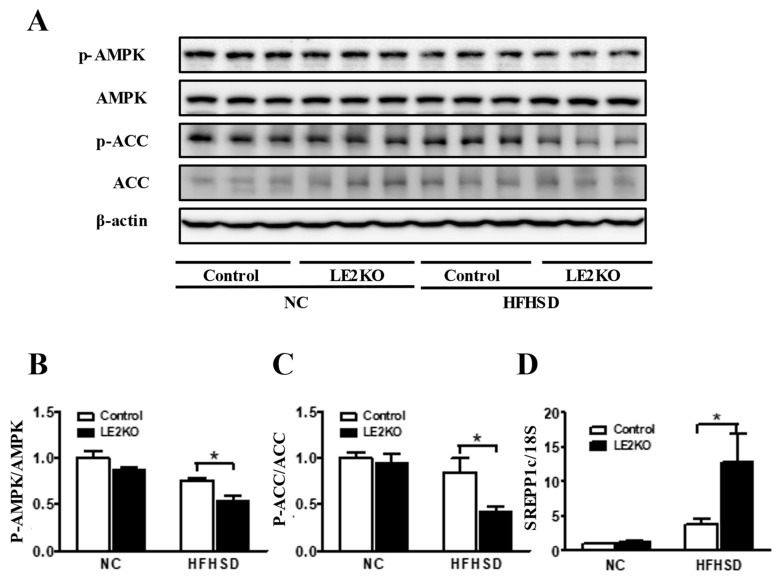
HFHSD-LE2KO decreased phosphorylation of AMPK/ACC and increased mRNA expression of SREBP-1c in liver. (**A**) Phosphorylation of AMPK (Thr172) and ACC (Ser 79) in the livers of controls and LE2KO on NC or HFHSD, respectively (*n* = 3 for each group). (**B**,**C**) Quantification of phosphorylation of AMPK/AMPK and phosphorylation of ACC/ACC. (**D**) mRNA expression levels of SREBP1c in the livers of control and LE2KO on NC or HFHSD, respectively (*n* = 6 for each group). Error bars represent SEM; *p* values were determined by ANOVA (* *p* < 0.05, vs. HFHSD-Control).

**Figure 3 ijms-23-08521-f003:**
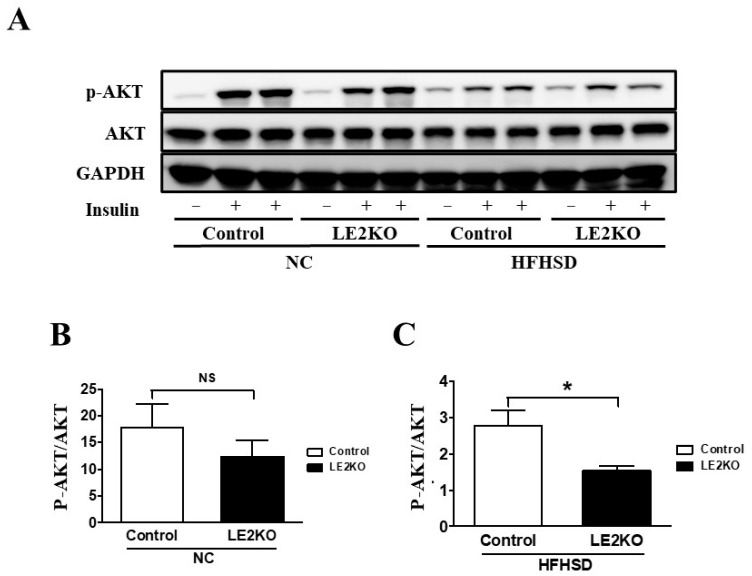
HFHSD-LE2KO have impaired insulin sensitivity in skeletal muscle. (**A**) Phosphorylation of AKT (Ser 473) in the soleus muscle of controls and LE2KO on NC or HFHSD, respectively (*n* = 3 for each group). (**B**,**C**) Quantification of phosphorylation of AKT/AKT on NC or HFHSD. Error bars represent SEM; *p* values were determined by *t*-test (* *p* < 0.05, vs. control).

**Figure 4 ijms-23-08521-f004:**
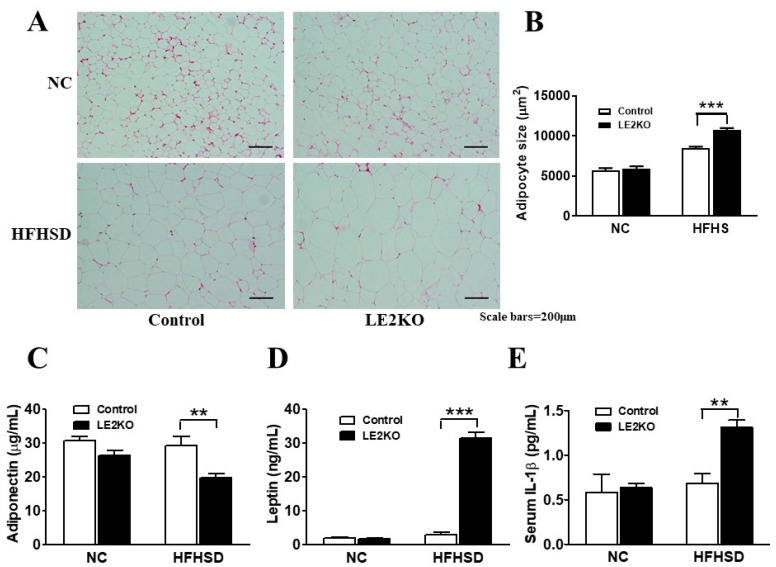
HFHSD-LE2KO display enlargement of adipocytes and changes in serum adipocytokines. (**A**) Typical microscopic images of adipocytes and (**B**) quantification of size of adipocyte (*n* = 8 for each group). (**C**–**E**) Serum adiponectin levels were lower and serum leptin and IL-1β levels were markedly higher in HFHSD-LE2KO than HFHSD-Control. (*n* = 8 for each group). Error bars represent SEM; *p* values were determined by ANOVA (** *p* < 0.01, *** *p* < 0.001 vs. HFHSD-Control).

**Figure 5 ijms-23-08521-f005:**
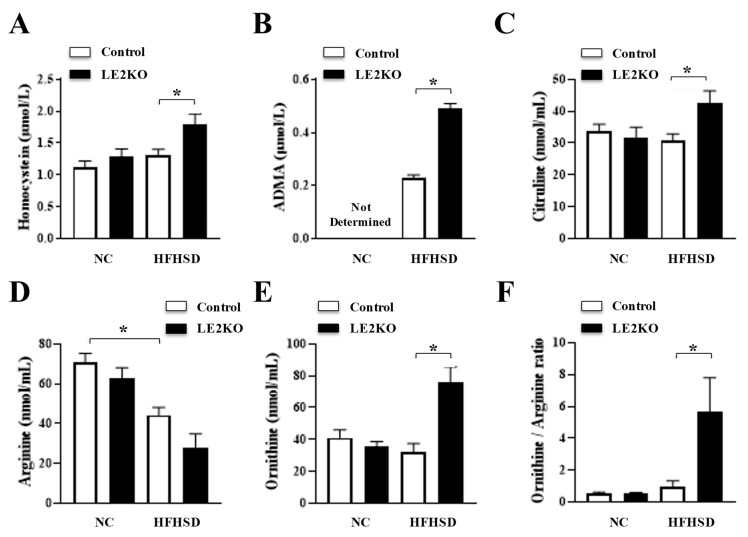
Serum Amino Acid Profile. (**A**) Homocystein, (**B**) asymmetric dimethylarginine (ADMA), (**C**) Citrulin levels were higher in HFHSD-LE2KO than HFHSD-Control. (**D**) Arginine level was lower in HFHSD-Control than NC-Control, and (**E**) Ornithine and (**F**) Ornithine/Arginine ratio were higher in HFHSD-LE2KO than HFHSD-Control. (*n* = 8 for each group). Data are mean ± SEM and median (interquartile range). (nmol/mL) *p* values were determined by ANOVA (* *p* < 0.05). *n* = 8 for each group.

**Figure 6 ijms-23-08521-f006:**
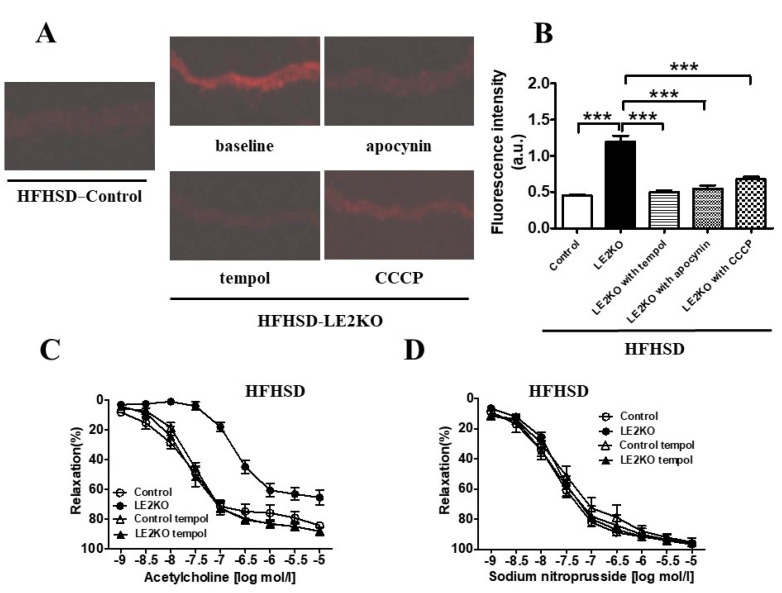
HFHSD-LE2KO demonstrate increased aortic superoxide levels and impaired endothelium-dependent relaxation. (**A**) DHE staining of aorta from HFHFD-Control, HFHSD-LE2KO, and HFHSD-LE2KO with pre-incubated of tempol, apocynin, and CCCP and (**B**) quantification of DHE fluorescence intensity (*n* = 5–6 for each group). Vascular relaxation of aortic rings with ACh (**C**) and SNP (**D**) on NC or HFHSD for 20 weeks, respectively (*n* = 8–10 for each group). Results of relaxation are expressed as percentage changes in steady-state level of contraction with 10^−6.5^ M phenylephrine. Error bars represent SEM; *p* values were determined by ANOVA (*** *p* < 0.001 vs. HFHSD-LE2KO without pre-incubation).

**Table 1 ijms-23-08521-t001:** Serum Fatty Acid Profile.

	NC		HFHSD		*p* Value
	Control	LE2KO	Control	LE2KO	
Body weight	31.4 ± 0.7	31.0 ± 0.8	41.5 ± 0.9 *	41.6 ± 0.9	<0.0001
Myristic acid	11.7 ± 3.8	16.2 ± 8.8	9.7 ± 1.1	7.4 ± 1.0 ^#^	0.03
Myristoleic acid	0.7 ± 0.4	1.1 ± 0.9	1.3 ± 0.3	1.2 ± 0.8	0.28
Palmitic acid	677.1 ± 64.1	840.0 ± 180.0	694.7 ± 88.8	756.1 ± 122.2	0.05
Palmitoleic acid	63.6 ± 11.8	86.3 ± 22.8	76.5 ± 12.2	57.2 ± 6.9 ^#^	0.005
ThankStearic acid	183.4 ± 18.7	221.0 ± 37.8	215.2 ± 17.0	353.3 ± 73.1^#^	<0.0001
Oleic acid	371.7 ± 50.2	519.4 ± 194.0	556.6 ± 96.4 *	676.2 ± 85.6	0.0003
Linoleic acid	792.8 ± 72.6	972.8 ± 284.1	633.7 ± 71.0	458.1 ± 83.4	<0.0001
γ-Linolenic acid	18.0 ± 2.9	22.9 ± 8.6	14.3 ± 1.5	9.7 ± 1.5 ^#^	0.002
α-Linolenic acid	9.9 ± 2.5	14.3 ± 7.8	6.7 ± 1.2 *	2.6 ± 0.6 ^#^	0.002
Arachidic acid	2.6 ± 0.4	3.3 ± 1.0	3.8 ± 0.6	5.3 ± 1.4 ^#^	<0.0001
Eicosenoic acid	7.4 ± 2.0	10.3 ± 5.8	8.3 ± 1.2	9.4 ± 2.3	0.35
Eicosadienoic acid	3.2 ± 1.0	4.2 ± 1.9	3.0 ± 0.8	3.7 ± 1.1	0.31
5-8-11 Eicosatetraenoic acid	2.2 ± 0.4	2.9 ± 0.8	5.2 ± 0.9 *	21.3 ± 9.7 ^#^	<0.0001
Dihomo-γ-linolenic acid	19.7 ± 2.4	25.9 ± 5.5	27.7 ± 5.7 *	73.4 ± 28.5 ^#^	0.0003
Arachidonic acid	310.3 ± 40.0	346.3 ± 50.5	488.5 ± 53.4 *	992.2 ± 282.1 ^#^	<0.0001
Eicosapentaenoic acid	30.6 ± 2.8	39.2 ± 10.4	19.4 ± 2.9 *	8.7 ± 2.9 ^#^	<0.0001
Behenic acid	6.7 ± 0.7	7.0 ± 0.8	9.7 ± 1.4 *	12.8 ± 3.5	0.0002
Erucic acid	1.9 ± 0.1	1.2 ± 0.4	1.6 ± 0.3	2.2 ± 0.6 ^#^	<0.0001
Docosatetraenoic acid	2.9 ± 0.5	1.9 ± 1.5	4.7 ± 0.9	9.4 ± 2.4 ^#^	<0.0001
Docosapentaenoic acid	11.9 ± 1.6	15.2 ± 3.2	9.8 ± 1.8	7.6 ± 1.2	<0.0001
Lignoceric acid	5.5 ± 0.5	5.8 ± 0.6	5.8 ± 0.5	6.6 ± 1.6	0.18
Docosahexaenoic acid	225.4 ± 25.1	266.9 ± 32.1	230.1 ± 21.5	271.9 ± 40.0 ^#^	0.006
Nervonic acid	15.9 ± 1.1	17.4 ± 2.0	21.4 ± 1.4 *	32.8 ± 4.9 ^#^	<0.0001
EPA/AA ratio	30.6 ± 2.8	39.2 ± 10.4	19.4 ± 2.9 *	8.7 ± 1.1 ^#^	<0.0001

Data are mean ± SD. (μg/mL). Final column reflects overall group differences. * *p* < 0.05 vs. NC- Control; ^#^ *p* < 0.05 vs. HFHSD-Control. *n* = 8 in all groups.

## Data Availability

Data supporting reported results can be obtained from the corresponding author under reasonable request.

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
