# Peer review of "Metabolic Remodeling with Hepatosteatosis Induced Vascular Oxidative Stress in Hepatic ERK2 Deficiency Mice with High Fat Diets"

_ijms, 2022, doi:10.3390/ijms23158521_

Round 1
Reviewer 1 Report
In this study, the authors revealed changes in extrahepatic tissues, such as adipose and skeletal muscle of LE2KO mice on a HFHSD. The results are disjointed and appeared as a collage of observations. There is overreliance on observations from their previous publication (ref 7), to the point where rationale for the investigation is lacking. The observations did not address the primary aim of this study, at best the observation is correlative.
(1) Body weight was not reported in Table 1. It was reported in ref (7). Some data is essential for completion.
(2) section 2.2. It was not clear which enzymes in FAs synthesis were upregulated and FA oxidation were down-regulated, as the data were neither presented in ref 7 (published in 2013) nor available in any public databases. As the raw data of the microarray was not deposited in any public domain, it should be made available for independent validation. Further the analysis pipeline was done by a company (Toray), so the analysis pipeline is unclear.
(3) Line 90. The change in phosphorylated AMPK, while statistically significant, was rather marginal. To offer more confidence that the change was sufficient to elicit a physiological response, downstream mediators of p-AMPK should be examined.
(4) Line 95. While the authors have leveraged on the previous pathways analysis (again ref 7) to identify fatty acid synthesis/metabolism, the same analysis also was enriched for genes involved in glucogenesis, glycolysis and glycogen metabolism, i.e. among the top 10 upregulated enriched pathways. However, in this study, the authors concluded no significant change in expression of two genes associated with these pathways. Without the availability of raw data for independent validation, it was not convincing.
(4) Line 113. Was there difference in the p-AKT/AKT in the isolated skeletal muscle between NC and LE2KO (-Control and -HFHSD) at basal level, i.e. without insulin treatment? For that matter, what was the rationale to isolate and treat with exogenous insulin in culture? What was the p-AKT/AKT in skeletal muscle directly isolated from mice that already had difference in serum insulin (Table 2).
(5) Line 115. What was the expression of insulin receptor and its phospho-activation status in the liver and skeletal muscle? This should be examined.
(6) In contrast to the graph in Figure 4B, the images in Figure 4A showed that the adipocytes were larger in LE2KO-NC compared with Control-NC. The surface area of adipocyte differs drastically depending on the plane of section, as evident from the images. Please clarify how the values and SEM, which was unusually small, in Figure 4B were obtained. It was also unclear if the data represent mean or median.
(7) Line 137. There was no strong evidence that hypertrophy has occurred. Hyperplasia could have happened, especially with a larger adipocyte size.
(8) Section 2.7. The information was from previous reference (ref 7), which already showed elevated levels of Nox1 and Nox4. Therefore, a high aortic ROS is expected. What about immune cell infiltration? Is there any connection to the serum cytokine profile?
Author Response
Point to point rebuttal for reviewers’ comments
Reviewer 1
In this study, the authors revealed changes in extrahepatic tissues, such as adipose and skeletal muscle of LE2KO mice on a HFHSD. The results are disjointed and appeared as a collage of observations. There is overreliance on observations from their previous publication (ref 7), to the point where rationale for the investigation is lacking. The observations did not address the primary aim of this study, at best the observation is correlative.
Thank you very much for the comments. We agree that the study is rather an observatory. We have previously shown the endothelial dysfunction with hepatosteatosis in HFHSD-LE2KO. Endothelial dysfunction is prominent in type1 DM models with very high glucose (> 400mg/dl). The glucose elevation in HFHSD-LE2KO was modest (=200mg/dl); however, HFHSD-LE2KO revealed endothelial dysfunction with oxidative stress. Indeed, in HFHSD-Control, endothelial function was almost normal. Therefore, we seek the metabolic changes to induce endothelial dysfunction in this model. We showed elevated FFA, hyperinsulinemia, and changes in w3/w6 and ornithine/arginine ratio. These metabolic changes had not been shown in the previous manuscript. This study aims to assess the potential metabolic changes to induce endothelial dysfunction in HFHSD-LE2KO.
- Body weight was not reported in Table 1. It was reported in ref (7). Some data is essential for completion.
We put data in text
- section 2.2. It was not clear which enzymes in FAs synthesis were upregulated and FA oxidation were down-regulated, as the data were neither presented in ref 7 (published in 2013) nor available in any public databases. As the raw data of the microarray was not deposited in any public domain, it should be made available for independent validation. Further the analysis pipeline was done by a company (Toray), so the analysis pipeline is unclear.
Thank you very much for the suggestion. We added the up-regulation and down-regulation pathway/gene in supplemental file.
- Line 90. The change in phosphorylated AMPK, while statistically significant, was rather marginal. To offer more confidence that the change was sufficient to elicit a physiological response, downstream mediators of p-AMPK should be examined.
Thank you for the suggestion. We checked the p-ACC, a downstream of AMPK, which was decreased in the liver from HFHSD-LE2KO. We also added the sentence in Results section.
(4) Line 95. While the authors have leveraged on the previous pathways analysis (again ref 7) to identify fatty acid synthesis/metabolism, the same analysis also was enriched for genes involved in glucogenesis, glycolysis and glycogen metabolism, i.e. among the top 10 upregulated enriched pathways. However, in this study, the authors concluded no significant change in expression of two genes associated with these pathways. Without the availability of raw data for independent validation, it was not convincing.
Thank you very much for the important point. We have checked the expression of PEPCK and G6Pase with RT-PCR. The increased expression of them in HFHSD-LE2KO did not reach significance. However, in the pathway analysis, ten glycolysis/gluconeogenesis genes are down-regulated. We consider that the glyconeogenesis genes were not significantly increased, and other glucose metabolic enzymes were down-regulated. We are also attaching the results of gene chips.
- Line 113. Was there difference in the p-AKT/AKT in the isolated skeletal muscle between NC and LE2KO (-Control and -HFHSD) at basal level, i.e. without insulin treatment? For that matter, what was the rationale to isolate and treat with exogenous insulin in culture? What was the p-AKT/AKT in skeletal muscle directly isolated from mice that already had difference in serum insulin (Table 2).
As in method 4.5, we injected insulin via tail vein, and skeletal muscle was immediately frozen, and western blotting was done. Mice were anesthetized, and soleus muscles appeared by cutting skins. 0.75U/kg body weight of human regular insulin (Humulin R, Eli Lilly, Indianapolis, IN, USA) was injected from the caudal vein. After 15 minutes of injection, soleus muscles were cut and rapidly frozen with liquid nitrogen. pAKT was detected with western blotting.
We injected a relatively high insulin concentration (0.75U/kg); we considered that the basal insulin levels could be ignored because of relatively low levels.
(5) Line 115. What was the expression of insulin receptor and its phospho-activation status in the liver and skeletal muscle? This should be examined.
We are sorry, we cannot do them for the limited time to collect enough proteins. According to gene chip analysis, expression levels of insulin receptor, IRS1, and IRS2 were similar between HFHSD-Control and HFHSD-LE2KO.
(6) In contrast to the graph in Figure 4B, the images in Figure 4A showed that the adipocytes were larger in LE2KO-NC compared with Control-NC. The surface area of adipocyte differs drastically depending on the plane of section, as evident from the images. Please clarify how the values and SEM, which was unusually small, in Figure 4B were obtained. It was also unclear if the data represent mean or median.
Thank you for the suggestions. The data was mean +/-SEM. We indicated it in the figure legend. We did not find a difference in mean adipose size between NC-Control and NC-LE2KO. The deviation was a little increased in NC-LE2KO.
(7) Line 137. There was no strong evidence that hypertrophy has occurred. Hyperplasia could have happened, especially with a larger adipocyte size.
Thank you for the suggestions. We cannot determine whether the changes in adipocytes were hypertrophy or hyperplasia. We use the term, `enlargement of adipocytes’
(8) Section 2.7. The information was from previous reference (ref 7), which already showed elevated levels of Nox1 and Nox4. Therefore, a high aortic ROS is expected. What about immune cell infiltration? Is there any connection to the serum cytokine profile?
Thank you very much. We do not observe the macrophages in the aortic walls. We also observe the elevated ROS in perivascular adipocytes (PVA) with some infiltrations of leuckocytes. We are now investigating the role of PVA in endothelial dysfunction in Mets.

Reviewer 2 Report
The manuscript entitled ”Metabolic Remodeling with Hepatosteatosis Induced Vascular Oxidative Stress in Hepatic ERK2 Deficiency Mice with High Fat Diets” is characterizing metabolic complications in the liver, skeletal muscle, visceral fat, and aorta in high fat high sucrose diet fed mice with liver specific KO of Erk2. The main results show that liver specific Erk2 KO mice on high fat high sucrose diet develop hepatosteatosis, decreased insulin sensitivity in skeletal muscle, enlargement of adipocytes, increased adipocytokine release, and vascular oxidative stress. The characterization of the adipocytes and skeletal muscle in this context is new, while the liver and aorta data complements (and in some parts overlaps) data from a previous manuscript from this research team. The manuscript has a clear structure and is overall well written, but I have a few comments and points that could be clarified.
General comments:
1. 1. In the introduction it is clearly stated that this mouse model has been previously published and some of the main findings from the previous paper are written out. However, some data is repeated from the previous paper (for example liver weight, blood glucose, serum insulin, serum lipids etc.) but is presented in a way that you get the expression that this is novel data. My suggestion is to mention the data that has been repeated but focus the figures and tables on the novel data.
2. 2. All the data in this article is as far as I can see from male mice. Have you found similar phenotype in the female mice, or are they more protected against diet induced hepatosteaotosis in this model? I would appreciate if the reasoning behind choice of gender was mentioned in the manuscript.
3. 3. The figures show the number of subjects used for each test in the figure legend, while this information is missing in all four tables. Please add the numbers of subjects for the different groups in the tables (or table footnotes) for clarification.
4. 4. The data on the serum fatty acids are of great interest, but it would also be interesting to see if they mirror the fatty acid compostion of the liver as well, or at least discuss if there is a correlation between the liver and serum fatty acids. Another aspect which would be interesting to look into is if there is a different expression in the desaturases and elongases involved in the fatty acid biosynthesis chain for those fatty acids that differ between genotypes.
Specific comments:
1. 1. For clarification it would be good to add “in serum” in the titles of Table 3 and Table 4.
2. 2. The increased adipocyte size in gonadal fat is interesting, but I’m missing a description in the methods about how the size was measured. Was it manually measured from microscope photos with scale bars or automatically measured by a software? From the Figure 4B it looks like the area of the adipocytes has been measured. Since larger adipocytes are not always spherical but often elongated, as can be seen in Figure 4A, I wonder how the measurement was done and if for example the average diameter of 3 measurements per adipocyte could be a more accurate measurement than the area (if manually measured)?
3. 3. In results section Line 70, it says that the “LE2KO and controls were weaned onto either HFHSD or normal chow (NC) for 20 weeks.” but in the methods section Line 141 it says “Eight-week-old male LE2KO and control littermates were fed with either NC or HFHSD… for 20 weeks”. Usually mice are weaned at an earlier age than 8 weeks (usually around 4 weeks of age) so I guess one of the sentences are incorrect and should be corrected.
4. 4. In methods section Line 177, it says that “Total RNA was isolated from livers and aorta” but in the manuscript I could only find RNA data from liver. Please correct.
Author Response
Point to point rebuttal for reviewers’ comments
Reviewer 2
The manuscript entitled ”Metabolic Remodeling with Hepatosteatosis Induced Vascular Oxidative Stress in Hepatic ERK2 Deficiency Mice with High Fat Diets” is characterizing metabolic complications in the liver, skeletal muscle, visceral fat, and aorta in high fat high sucrose diet fed mice with liver specific KO of Erk2. The main results show that liver specific Erk2 KO mice on high fat high sucrose diet develop hepatosteatosis, decreased insulin sensitivity in skeletal muscle, enlargement of adipocytes, increased adipocytokine release, and vascular oxidative stress. The characterization of the adipocytes and skeletal muscle in this context is new, while the liver and aorta data complements (and in some parts overlaps) data from a previous manuscript from this research team. The manuscript has a clear structure and is overall well written, but I have a few comments and points that could be clarified.
General comments:
- In the introduction it is clearly stated that this mouse model has been previously published and some of the main findings from the previous paper are written out. However, some data is repeated from the previous paper (for example liver weight, blood glucose, serum insulin, serum lipids etc.) but is presented in a way that you get the expression that this is novel data. My suggestion is to mention the data that has been repeated but focus the figures and tables on the novel data.
Thank you very much. We corrected them following your suggestions. According to the suggestions, we widely revised introduction and the first paragraph of discussion.
- All the data in this article is as far as I can see from male mice. Have you found similar phenotype in the female mice, or are they more protected against diet induced hepatosteaotosis in this model? I would appreciate if the reasoning behind choice of gender was mentioned in the manuscript.
Thank you very much for the important comments. Unfortunately, we did not observe hepatosteatosis in female mice. It is generally hard to induce endothelial dysfunction in any disease model except for ovariectomy or very old ages. Thus, we chose male mice for the study.
- The figures show the number of subjects used for each test in the figure legend, while this information is missing in all four tables. Please add the numbers of subjects for the different groups in the tables (or table footnotes) for clarification.
Thank you very much. n=8, We corrected as you suggested.
- The data on the serum fatty acids are of great interest, but it would also be interesting to see if they mirror the fatty acid compostion of the liver as well, or at least discuss if there is a correlation between the liver and serum fatty acids. Another aspect which would be interesting to look into is if there is a different expression in the desaturases and elongases involved in the fatty acid biosynthesis chain for those fatty acids that differ between genotypes.
This is a very important comment. Unfortunately, we do not have data about the FFA in the liver. We add the discussion about the acceleration of FFA synthesis in the liver.
Specific comments:
- For clarification it would be good to add “in serum” in the titles of Table 3 and Table 4.
Thank you. We added them.
- The increased adipocyte size in gonadal fat is interesting, but I’m missing a description in the methods about how the size was measured. Was it manually measured from microscope photos with scale bars or automatically measured by a software? From the Figure 4Bit looks like the area of the adipocytes has been measured. Since larger adipocytes are not always spherical but often elongated, as can be seen in Figure 4A, I wonder how the measurement was done and if for example the average diameter of 3 measurements per adipocyte could be a more accurate measurement than the area (if manually measured)?
We thank the reviewer for this comment and apologize for the lack of explanation. We automatically measured adipocyte size from microscope photos with a software of BZ-8000 (Keyence, Osaka, Japan). In response, we have now clarified this point in the revised manuscript.
- In results section Line 70, it says that the “LE2KO and controls were either HFHSD or normal chow (NC) for 20 weeks.” but in the methods section Line 141it says “Eight-week-old male LE2KO and control littermates were fed with either NC or HFHSD… for 20 weeks”. Usually mice are weaned at an earlier age than 8 weeks (usually around 4 weeks of age) so I guess one of the sentences are incorrect and should be corrected.
Thank you for this comment and we apologize for the confusion. “Eight-week-old male LE2KO and control littermates were fed with either NC or HFHSD for 20 weeks.” is right. In response to the comment, we have revised this sentence in Results section.
- In methods sectionLine 177, it says that “Total RNA was isolated from livers and aorta” but in the manuscript I could only find RNA data from liver. Please correct.
Thank you, Corrected.

Reviewer 3 Report
1- is there any conclusion for the study?
2-please briefly explain the previous results you mentioned in the first paragraph of discussion
3-the main findings of the paper is better to be mentioned at the beginning of the Discussion and discussed further
4-please write the references before the . in every sentence
5- Can you summarize table 4 and put the unnecessary info in the supplenemtary file?
6- can you make a graph out of table 4 to see the importance of the p values?
7- Figure 6 is not sited in the text
8- what is the importance of Figure 6?
9-
Author Response
Point to point rebuttal for reviewers’ comments
Reviewer 3
- is there any conclusion for the study?
- please briefly explain the previous results you mentioned in the first paragraph of discussion
- the main findings of the paper is better to be mentioned at the beginning of the Discussion and discussed further
Thank you very much for the comments. We agree to change some structures of the manuscript. We have previously shown the endothelial dysfunction with hepatosteatosis in HFHSD-LE2KO. Endothelial dysfunction is prominent in type1 DM models with very high glucose (> 400mg/dl). The glucose elevation in HFHSD-LE2KO was modest (=200mg/dl); however, HFHSD-LE2KO revealed endothelial dysfunction with oxidative stress. We also showed elevated FFA and hyperinsulinemia. The previous manuscript has not shown the potential metabolic changes to induce endothelial dysfunction in HFHSD-LE2KO. This study aimed to assess the potential metabolic changes to induce endothelial dysfunction in HFHSD-LE2KO. According to the suggestions, we widely revised introduction and the first paragraph of discussion.
- please write the references before the . in every sentence
Thank you very much. We correct them.
- Can you summarize table 4 and put the unnecessary info in the supplenemtary file?
We have summarized Table 4 to the graph in Figure 5 and added them in the supplemental file.
6-can you make a graph out of table 4 to see the importance of the p values?
Thank you for the suggestion. The important points of amino acids analysis were changes in arginine metabolisms and homocysteine because they could be related to endothelial dysfunction. We moved table to supplement and show the points in the graph (Figure 5)
7- Figure 6 is not sited in the text
8- what is the importance of Figure 6?
Thank you for suggestions. Fig 6 was graphic abstract of this manuscript that is requesting for publication.
Round 2
Reviewer 1 Report
The authors have attempted to address most of my comments.
The excel file containing the up- and down regulated genes should be made available as supplemental data to this manuscript. Similar data on Up and down gene Control-LE2KO should also be included.
Author Response
Point to point rebuttal for reviewers’ comments
Reviewer 1
The authors have attempted to address most of my comments.
We appreciate the comment.
The excel file containing the up- and down regulated genes should be made available as supplemental data to this manuscript. Similar data on Up and down gene Control-LE2KO should also be included.
Thank you for the comment. We add the comparison between NC-Control vs HFHSD-Control and add the gene chip data in supplemental data,

Reviewer 3 Report
Dear Authors
Thank you for your reply,
The manuscript had improved a lot
thank you
Author Response
Point to point rebuttal for reviewers’ comments
Reviewer 3
Thank you for your reply,
The manuscript had improved a lot thank you
We appreciate the comment, which improves the manuscript a lot.